# Naked *d*-orbital in a centrochiral Ni(II) complex as a catalyst for asymmetric [3 + 2] cycloaddition

Yoshihiro Sohtome[1,2], Genta Nakamura[1,3], Atsuya Muranaka[2,4], Daisuke Hashizume[5], Sylvain Lectard[1], Teruhisa Tsuchimoto[3], Masanobu Uchiyama[2,4] & Mikiko Sodeoka[1,2]

Chiral metal catalysts have been widely applied to asymmetric transformations. However, the electronic structure of the catalyst and how it contributes to the activation of the substrate is seldom investigated. Here, we report an empirical approach for providing insights into the catalytic activation process in the distorted Ni(II)-catalysed asymmetric [3 + 2] cycloaddition of α-ketoesters. We quantitatively characterize the bonding nature of the catalyst by means of electron density distribution analysis, showing that the distortion around the Ni(II) centre makes the $dz^2$ orbital partially 'naked', wherein the labile acetate ligand is coordinated with electrostatic interaction. The electron-deficient $dz^2$ orbital and the acetate act together to deprotonate the α-ketoester, generating the ($\Lambda$)-Ni(II)–enolate. The solid and solution state analyses, together with theoretical calculations, strongly link the electronic structure of the centrochiral octahedral Ni(II) complex and its catalytic activity, depicting a cooperative mechanism of enolate binding and outer sphere hydrogen-bonding activation.

[1] Synthetic Organic Chemistry Laboratory, RIKEN, Wako, Saitama, 351-0198, Japan. [2] RIKEN Center for Sustainable Resource Science, Wako, Saitama, 351-0198, Japan. [3] Department of Applied Chemistry, School of Science and Technology, Meiji University, Kawasaki, Kanagawa, 214-8571, Japan. [4] Elements Chemistry Laboratory, RIKEN, Wako, Saitama, 351-0198, Japan. [5] RIKEN Center for Emergent Matter Science, Wako, Saitama, 351-0198, Japan. Correspondence and requests for materials should be addressed to Y.S. (email: sohtome@riken.jp) or to D.H. (email: hashi@riken.jp) or to M.S. (email: sodeoka@riken.jp).

Optimization in asymmetric metal-based catalysis generally requires trial-and-error approaches, whereby a metal source and a 'privileged' chiral ligand[1] are combined without understanding of the three-dimensional (3D) and electronic structure of the active catalyst. Considering that synergistic activation of two reaction components is a key mechanistic strategy to attain high rate acceleration and selectivity[2,3], it is particularly important to understand the electron density distribution (EDD) of the chiral catalyst, to gain insight into how the metal and ligand(s) cooperatively activate two distinct reaction components.

Over recent decades, the use of asymmetric nickel catalysts has substantially evolved both in acid–base and redox catalysis[4–6]. Several examples of chiral nickel complexes that act as asymmetric catalysts have been sporadically characterized by X-ray structure analysis, opening up a window of opportunity to discuss the stereo-discrimination process in asymmetric nickel catalysis[7–17]. Evans's transition-state model, which was proposed on the basis of X-ray analysis of the Ni(II)–diamine–enolate complex, is a remarkable example, explaining the stereochemical course in the catalytic asymmetric Michael reaction of nitroolefins with 1,3-dicarbonyl compounds[7,8]. However, despite the rapid progress in this field, the electronic structure of the asymmetric nickel catalyst is seldom investigated and exploited. The difficulties in understanding nickel asymmetric catalysts arise from several features: (i) They exhibit various oxidation states (0, +1, +2, +3 and +4) and electronic configurations (typically from 16e to 20e). (ii) The steric factor and the coordination numbers of the ligand(s) drastically influence the coordination geometry and the assembly-state of nickel complexes. (iii) The structure changes dynamically depend on solvents, substrates and additives due to the coordination equilibrium[18].

Here, we describe the discovery and characterization of a distorted chiral Ni(II)–diamine–acetate complex and link the electronic structure to its catalytic activity (Fig. 1). In contrast to the classical symmetric octahedral Ni(II) catalysts (Fig. 1a), the Ni(II) catalyst reported here exhibits ($\Lambda$)-chirality at the Ni(II) center[19–23], in which the coordination about the nickel atom is a distorted octahedron (Fig. 1b). The distortion in the octahedral Ni(II) complex makes the $dz^2$ orbital partially

'naked' due to the labile acetate ligand, which is weakly coordinated by electrostatic interaction. This unique electronic feature, involving both Lewis acidic and Brønsted basic natures, facilitates enolization of the $\alpha$-ketoester. The distorted Ni(II) complex also exhibits hydrogen bond donor ability of the coordinated amine ligand, providing an outersphere binding site for the approaching electrophile[24]. With the bifunctional catalytic motif on the ($\Lambda$)-Ni(II) complex, we uncover the reactivity of the Ni(II)–enolate as a formal 1,3-dipolarophile that promotes [3 + 2] nitrone cycloaddition. The regioselectivity for this transformation is distinct from that of classical [3 + 2] cycloadditions[25]; the vast majority of nitrone cycloadditions are predicated on the use of electron-deficient alkenes (Fig. 1c). This report provides insights into the elusive mechanistic basis for inverse-electron-demand (IED) [3 + 2] cycloaddition[25–34] of $\alpha$-ketoesters with ($E$)-nitrones (Fig. 1d).

## Results

**Discovery of catalytic triad.** Based on our Ni(II)–enolate chemistry[35], we initiated this study by exploring a new Ni(II) complex, focusing on formal [3 + 2] cycloaddition[25] of $\alpha$-ketoesters **1** with nitrones **2** for the following reasons. (i) Control of the ambiphilic reactivities of $\alpha$-ketoesters, which exhibit both nucleophilic and electrophilic natures, constitutes a general challenge[36,37]. (ii) Catalytic asymmetric [3 + 2] cycloaddition using transient enolate has yet to be developed, despite the long history of enolate studies[25,38,39]. Only a few IED cycloadditions with electron-rich alkenes (such as enol ethers and silyl ethers), not enolate, have been reported with asymmetric catalysts[25–34]. The [3 + 2] cycloaddition, which is controlled by the lowest unoccupied molecular orbital of the dipole and the highest occupied molecular orbital of the 1,3-dipolarophile, can be also categorized as 'type III' according to Sustmann's classification[40]. (iii) The proposed catalytic reaction would provide stereochemically complex isoxazolidines **3** that bear three contiguous stereocenters, including a unique, stereochemically defined, hemiketal moiety.

We investigated the proposed formal [3 + 2] cycloaddition using cyclic nitrones **2** to construct chiral tetrahydroisoquinolines, which are one of the privileged scaffolds for drug

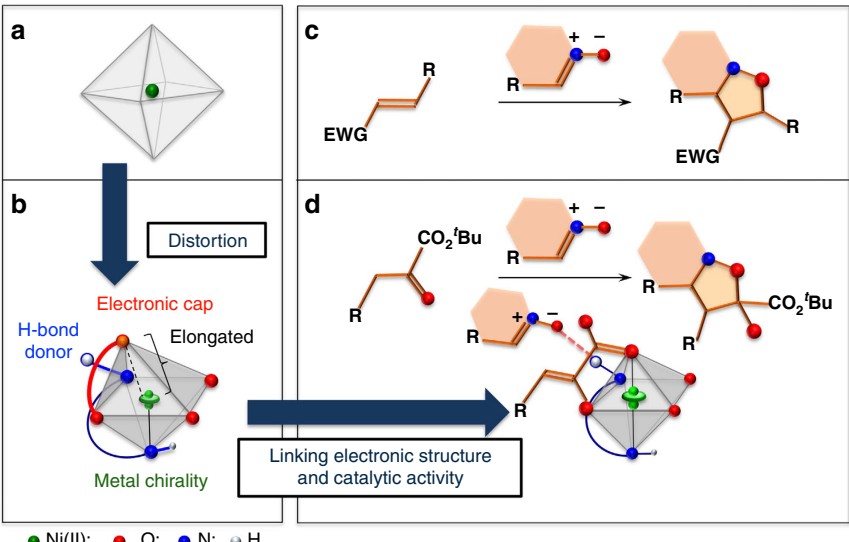

**Figure 1 | Linking the distorted Ni(II) complex to catalytic activity for inverse-electron-demand [3 + 2] cycloaddition.** (**a**) Usual symmetric octahedral Ni(II), (**b**) distorted octahedral Ni(II) complex, which enables acid–base catalysis on the metal centrochirality, (**c**) normal electron-demand [3 + 2] cycloaddition and (**d**) IED cycloaddition. EWG, electron-withdrawing group.

**Table 1 | Optimization of the catalytic system for formal [3 + 2] cycloaddition of α-ketoeseter 1a with (E)-nitrone 2a.**

| Entry | Metal source | (R,R)-diamine 4R | Additive (x mol%) | Yield (%)* | d.r.† | e.e. (%) |
|---|---|---|---|---|---|---|
| 1 | Ni(OAc)$_2$·4H$_2$O | 4a: benzyl | — | 91 | 20/1 | 50 |
| 2 | Cu(OAc)$_2$·2H$_2$O | 4a: benzyl | — | Trace | ND | ND |
| 3 | Zn(OAc)$_2$·2H$_2$O | 4a: benzyl | — | 19 | 3/1 | 70 |
| 4 | Pd(OAc)$_2$ | 4a: benzyl | — | Trace | ND | ND |
| 5 | Ni(octanoate)$_2$·nH$_2$O | 4a: benzyl | — | 87 | 20/1 | 44 |
| 6 | Ni(benzonoate)$_2$·nH$_2$O | 4a: benzyl | — | 72 | 20/1 | 44 |
| 7 | NiBr$_2$ | 4a: benzyl | — | 63 | 15/1 | 24 |
| 8 | Ni(OAc)$_2$·4H$_2$O | 4b: isobutyl | — | 88 | 30/1 | 40 |
| 9 | Ni(OAc)$_2$·4H$_2$O | 4c: isopropyl | — | 94 | 20/1 | 64 |
| 10 | Ni(OAc)$_2$·4H$_2$O | 4d: 3-pentyl | — | 83 | >50/1 | 61 |
| 11 | Ni(OAc)$_2$·4H$_2$O | 4e: cyclohexyl | — | 88 | >50/1 | 83 |
| 12‡ | Ni(OAc)$_2$·4H$_2$O | 4e: cyclohexyl | Et$_3$N (10) | 88 | >50/1 | 83 |
| 13‡ | Ni(OAc)$_2$·4H$_2$O | 4e: cyclohexyl | $^i$Pr$_2$NH (10) | 89 | >50/1 | 91 |
| 14§ | Ni(OAc)$_2$·4H$_2$O | 4e: cyclohexyl | $^i$Pr$_2$NH (10) | 71 | >50/1 | 91 |

*Yields are for isolated materials.
†d.r. Values were determined from the $^1$H NMR spectra of crude mixture.
‡1.2 equiv. of 2a was used.
§5 mol% of metal complexes and 1.2 equiv. of 2a were used. ND; not determined.

discovery[41]. Among the reaction parameters we investigated, the effect of metal was most drastic (Table 1, entries 1–4). For example, Ni(OAc)$_2$·4H$_2$O–(R,R)-4a complex[35] selectively produced *anti,anti*-3aa as the major product in 91% yield, 50% enantiomeric excess (e.e.) with 20/1 diastereomeric ratio (d.r.), while other metal acetate complexes displayed lower reactivity (Table 1, entries 1–4). As the counteranion, acetate gave the best results (Table 1, entries 5–7). The NiBr$_2$–(R,R)-4a complex developed by Evans[7,8] was less effective in this reaction (Table 1, entry 7). Systematic variation of the diamine ligand 4 revealed that α-branched substituents on the amine in (R,R)-4 are crucial for improving the enantioselectivity (Table 1, entries 8–11). The diamine (R,R)-4e, bearing cyclohexyl groups, was most effective, raising the e.e. value of 3aa to 83% e.e. Intriguingly, the diamine 4e has been overlooked as a chiral ligand for over a decade despite its potential utility and simplicity[42]. We finally found that diisopropylamine ($^i$Pr$_2$NH: 10 mol%) can further improve not only the reactivity, but also the enantioselectivity[43], affording (1R,2R,10bS)-3aa in 71% yield, >50/1 d.r. and 91% e.e., with 5 mol% of Ni(OAc)$_2$·4H$_2$O–(R,R)-4e (Table 1, entry 14). These results suggest that achiral $^i$Pr$_2$NH may participate in tuning the chiral environments constructed by the catalyst, without interrupting enolate formation by the Ni(II) catalyst.

**Structural determination of Ni(II) complexes in the solid state.** Removing the solvent of Ni(OAc)$_2$·4H$_2$O/(1R,2R)-4e = 1/1 mixture in tetrahydrofuran (THF) and subsequent crystallization from CH$_2$Cl$_2$ at −20 °C gave the rich green-colored I in 85% yield, while crystallization from n-hexane at room temperature afforded the light green-colored II in 86% yield. Both products are easily handled, air-stable complexes (Fig. 2a).

The X-ray structure analyses unambiguously revealed that I is a mononuclear Ni(II) complex: (Λ)-Ni(OAc)$_2$4e, while II is a μ-acetate-bridged trinuclear Ni(II) complex: (Λ,Λ)-[Ni$_3$ (μ-OAc)$_4$(OAc)$_2$4e$_2$] (Fig. 2b). In contrast to chiral nickel(II)–diamine catalysts identified so far[7–11] (Supplementary Figs 3–5), the newly developed complexes are chiral-at-Ni(II) center[19–23], in which the initial $C_2$-symmetry of the ligands on I and II is desymmetrized through formation of the Ni(II) complex. Mononuclear Ni(II) complex I has a distorted octahedral architecture; the atomic distance between Ni(II) and N(2) (2.109(1) Å) is longer than that of Ni(II)–N(1) (2.091(1) Å), and the N(2)–Ni(II)–O(4) angle is 155.99(5)°, which differs markedly from 180°; we show the longer Ni(II)–N(2) bond in pseudoapical position in Fig. 2b, left side. The intermolecular H-bonding interaction of the N(1)–H with THF in the crystal is involved, indicating that N(1)–H can intermolecularly activate the Lewis basic substrate[24], nitrone 2 (refs 34,44,45), in the present catalysis. The coordination patterns of the acetate anions in I also have a characteristic role in constructing a specific chiral environment; one of the acetates coordinates to the Ni-centre in the equatorial-equatorial mode, in which the acetate anion is symmetrically bridging [C(19)–O(1): 1.265(2) Å, C(19)–O(2): 1.266(2) Å]. In contrast, the other acetate that occupies pseudoapical and equatorial positions take asymmetric form [C(21)–O(3): 1.271(2) Å, C(21)–O(4): 1.259(2) Å]. Another feature in the distorted octahedral complex I is the distance between Ni(II) and pseudoapical O(4) [2.302(2) Å], which is significantly longer than other Ni(II)–O and Ni(II)–N distances [2.045(1)–2.140(1) Å], suggesting weak coordination.

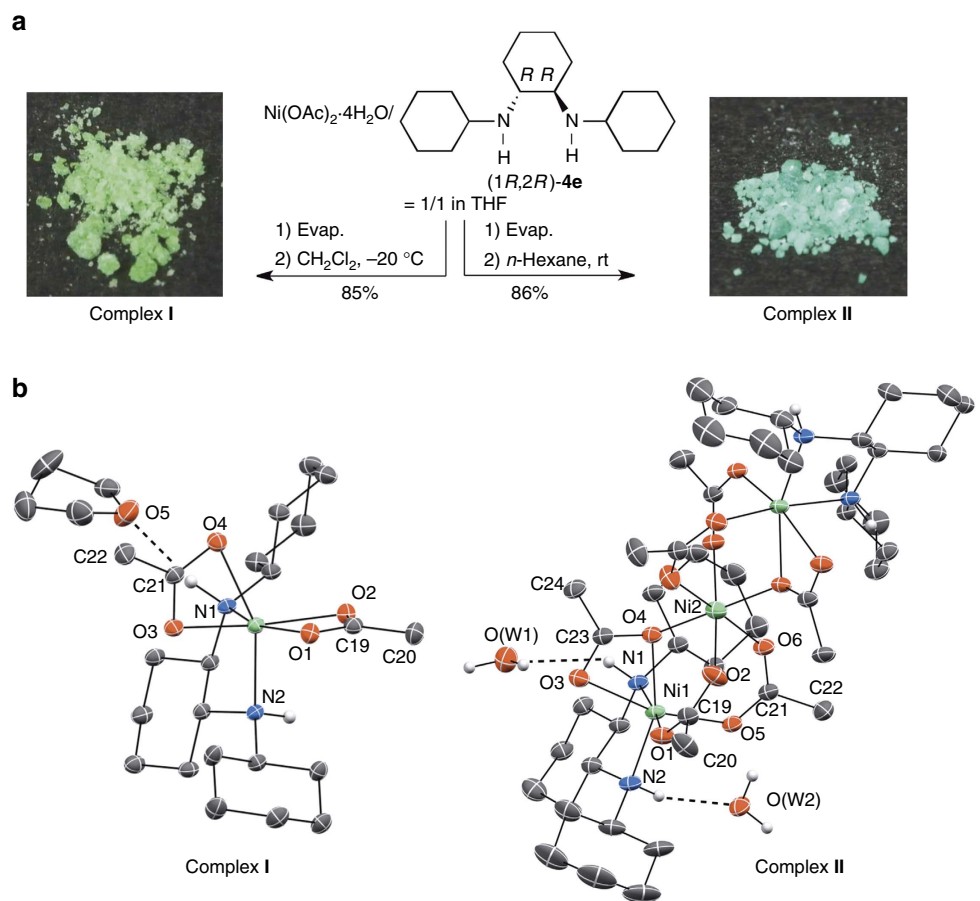

**Figure 2 | Structural determination of I and II in the solid state.** (**a**) Procedure for preparation of **I** and **II**. (**b**) ORTEP drawings of **I** and **II** (50% probability ellipsoids; hydrogen atoms on carbons in **I** and **II** and the minor disorder component of THF are omitted for the sake of clarity; hydrogen bonds are represented by broken lines).

**Table 2 | Catalytic activities of I and II.**

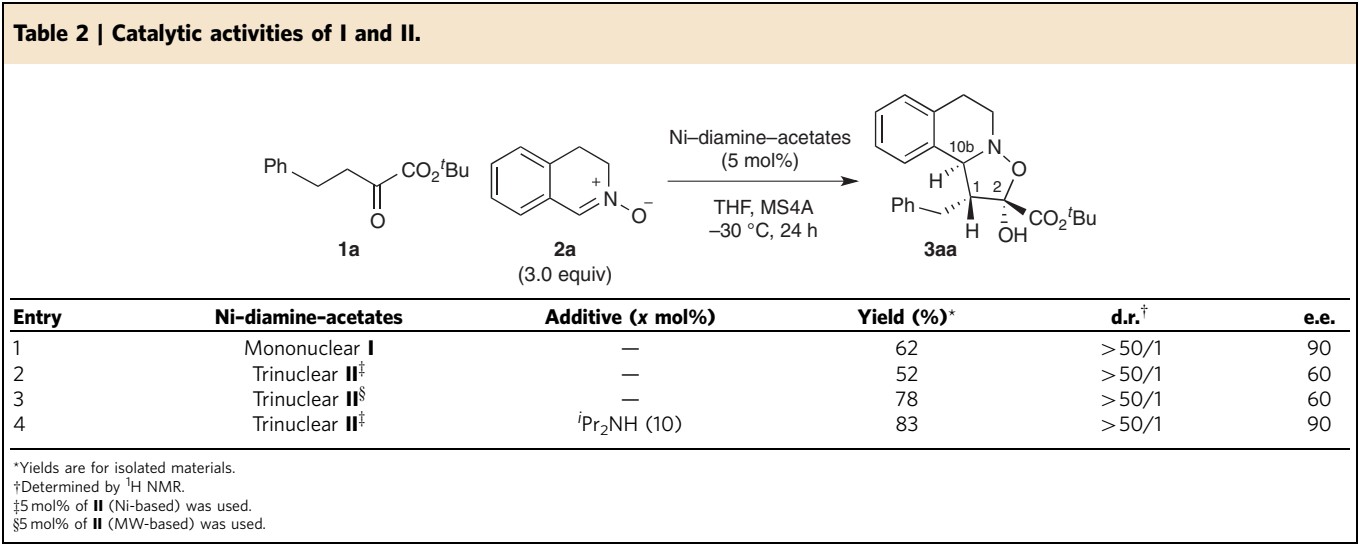

| Entry | Ni–diamine–acetates | Additive (x mol%) | Yield (%)* | d.r.† | e.e. |
|---|---|---|---|---|---|
| 1 | Mononuclear **I** | — | 62 | >50/1 | 90 |
| 2 | Trinuclear **II**‡ | — | 52 | >50/1 | 60 |
| 3 | Trinuclear **II**§ | — | 78 | >50/1 | 60 |
| 4 | Trinuclear **II**‡ | $^i$Pr$_2$NH (10) | 83 | >50/1 | 90 |

*Yields are for isolated materials.
†Determined by $^1$H NMR.
‡5 mol% of **II** (Ni-based) was used.
§5 mol% of **II** (MW-based) was used.

An obvious structural difference of the trinuclear Ni(II) complex **II** from the reported triple-bridged trinuclear Ni(II) complexes[46] is the distorted units at the terminal analogous to **I**, in which symmetric octahedral central nickel is linked with acetate bridges (Fig. 2b, right side). The angles of the bridging acetate [O(1)–C(19)–O(2): 127.40(18)°, O(5)–C(21)–O(6): 126.47(18)°] in complex **II** become larger than that of the corresponding acetate [O(1)–C(19)–O(2): 120.36(15)°] in **I**.

**Catalytic activities of I and II.** The investigation for catalytic activity of complexes **I** and **II** led to identification of two key

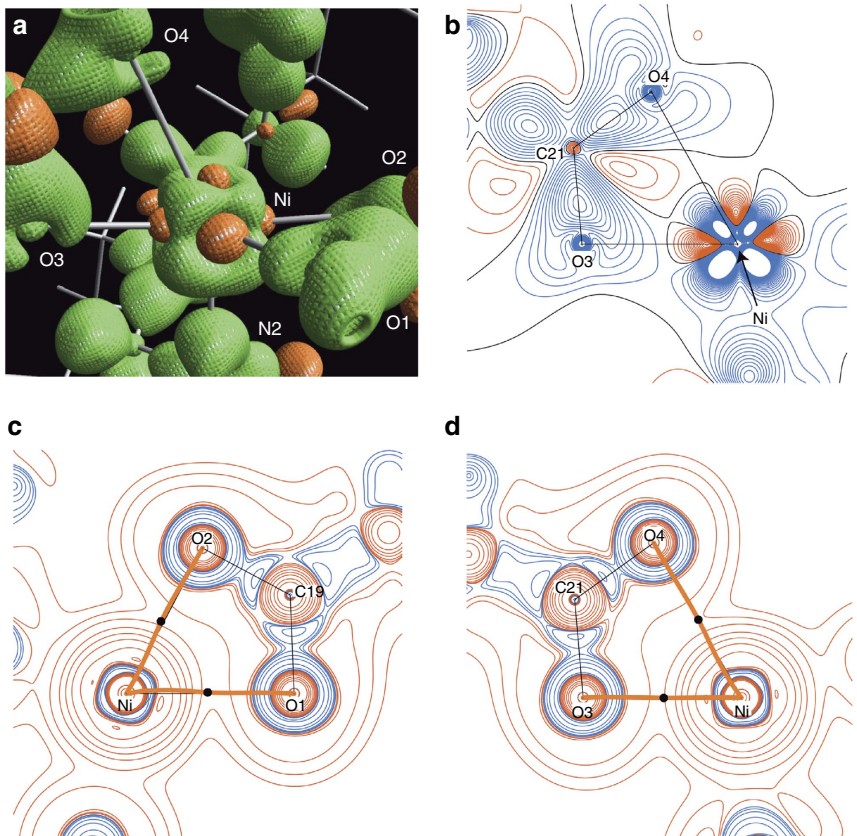

**Figure 3 | Electron density distribution maps of I.** (**a**) 3D isosurface static deformation density of **I**; surfaces drawn at $+0.2\,e\,Å^{-3}$ in green and at $-0.2\,e\,Å^{-3}$ in orange, (**b**) static model map on the O(3)–Ni–O(4) plane; contours drawn at $0.05\,e\,Å^{-3}$ interval in blue (positive), red (negative) and black (zero) lines, Laplacian distribution of total EDD (**c**) on the O(1)–Ni–O(2) plane, (**d**) on the O(3)–Ni–O(4) plane; the blue and red lines denote negative and positive Laplacian contours, respectively. The contours are drawn at $\pm 2 \times 10^n$, $\pm 4 \times 10^n$, $\pm 8 \times 10^n$ (where $n = 0, 1, 2$) $e\,Å^{-5}$. Bond path (BP) and bond critical points (BCPs) are depicted as orange lines and black dots, respectively, in **c,d**.

catalytic factors for attaining high enantioselectivity (Table 2). First, the mononuclear Ni(II) complex **I** provided better e.e. value than the trinuclear Ni(II) complex **II**. The mononuclear catalyst **I** produced the $[3+2]$ cycloaddition adduct **3aa** in 62% yield with $>50/1$ d.r. and 90% e.e., while the trinuclear catalyst **II** afforded **3aa** with poor enantioselectivity regardless of the amount of catalyst (entries 2 and 3). Second, addition of a catalytic amount of *achiral* $^i\mathrm{Pr_2NH}$ (10 mol%) improved the poor enantioselectivity with trinuclear catalyst **II** (Table 2, entry 2: 60% e.e. vs entry 4: 90% e.e.)[43].

**Electron density distribution analysis of I.** To characterize the bonding mode around the distorted Ni(II) in **I**, we performed EDD analysis[47] using single-crystal X-ray diffraction data (Fig. 3, Supplementary Data 1). A 3D plot of the static deformation density of **I** highlights its valence electron density topology (Fig. 3a). The distribution of $d$-orbitals along the coordination axes of the Ni(II) centre were found as electron-deficient regions. With respect to ligands, the lone pairs of oxygen [O(1), O(2) and O(3)] and nitrogen [N(1) and N(2)] atoms are directed to the electron-deficient regions at Ni(II), showing the conventional character of coordination bonds. In sharp contrast, the lone pair of O(4) is directed towards the electron-rich region at Ni(II). The 2D static model map on the O(3)–Ni–O(4) plane also demonstrates the different bonding character between the Ni–O(3) and the Ni–O(4) bonds (Fig. 3b). We can also discuss the unique bonding nature of Ni–O(4) with maps of

Laplacian distribution $[\nabla^2\rho(\mathbf{r})]$ of total EDD along bond paths by comparing the $\nabla^2\rho(\mathbf{r})$ distribution in the O(1)–Ni–O(2) plane (Fig. 3c) and the O(3)–Ni–O(4) plane (Fig. 3d)[48–50]. The valence shell charge concentration (VSCC) region at the Ni(II) centre are located between the bond paths of the Ni–O(1) and the Ni–O(2) bonds on the O(1)–Ni–O(2) plane (Fig. 3c). The VSCCs on O(1) and O(2) expand towards the charge-depletion regions around the Ni(II) centre along the bond paths. In contrast, the Ni–O(4) bond path goes through inside the VSCC region at O(4) and the VSCC region at the Ni(II) centre on the O(3)–Ni–O(4) plane (Fig. 3d). On the other hand, the Ni–O(3) bond shows similar features to the Ni–O(1) and Ni–O(2) bonds. The results described in Fig. 3 represent experimental evidence that a weaker orbital interaction between Ni(II) and O(4) is involved, while electrons are donated from lone pairs in the orbitals on other N and O atoms to the unfilled $d$-orbital on Ni(II) (Supplementary Figs 6–8). The density at the bond critical point (BCP) of the Ni(II)–O(4) bond $[0.260(2)\,e\,Å^{-3}]$ is remarkably lower than at the other coordination bonds: Ni(II)–O(1); 0.431(2), Ni(II)–O(2); 0.480(2), Ni(II)–O(3); 0.494(2), Ni(II)–N(1); 0.559(3), and Ni(II)–N(2); 0.550(3) $e\,Å^{-3}$. Thus, the dissymmetric, distorted octahedral Ni(II)–diamine–acetates **I** possessing an elongated Ni(II)–oxygen bond has $d^8$ 18-electronic configuration with a weak electrostatic interaction with O(4) at the pseudoapical position. The density at the BCP of the N(1)–H···O(5) is $0.084(10)\,e\,Å^{-3}$, which fits reasonably with the topological properties $[d(\mathrm{H}\cdots\mathrm{O})$: 2.200 Å, $d(\mathrm{N}\cdots\mathrm{O})$: 3.155(2) Å, $\alpha(\mathrm{N–H}\cdots\mathrm{O})$: 159.20°] of the H-bonding[51].

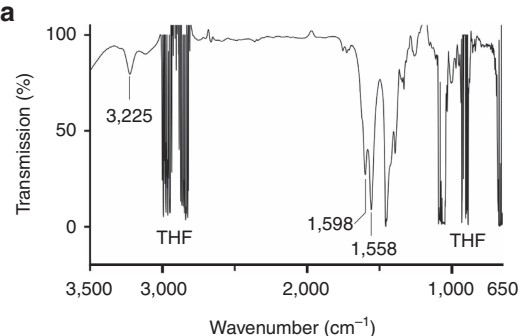

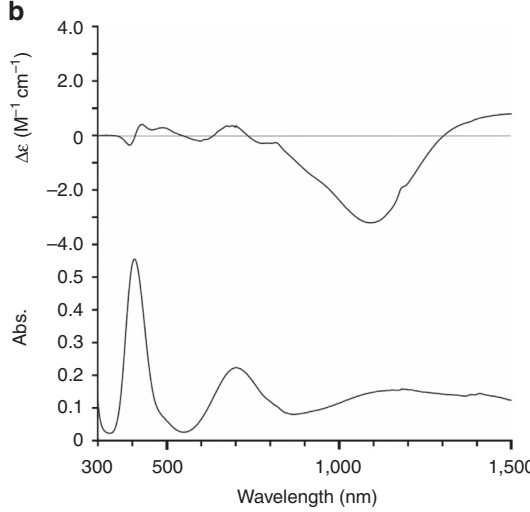

**Figure 4 | Structural analysis of I in solution.** (**a**) IR spectra of **I** in THF (0.017 M) and (**b**) electronic absorption and ECD spectra (300–1,500 nm) of **I** in THF (0.017 M).

**Structural analysis of I in solution.** We then characterized the structure of mononuclear Ni(II) complex **I** in THF (Fig. 4), at the same catalyst concentration as used for the IED [3 + 2] cycloaddition reaction. The unique shift for carboxylates (1,598 and 1,558 cm$^{-1}$) in the infrared spectrum supports the idea that the two distinct acetates coordinate to the Ni-centre in a bidentate manner (Fig. 4a)[52]. The band observed at 3,225 cm$^{-1}$ also suggests that the N–H functionality on the ligand can act as a proton donor in THF[53]. The electronic absorption spectrum of **I**, which shows broad bands at 408, 753 and 1,182 nm, is typical of pseudooctahedral geometry of Ni($d^8$) (Fig. 4b, bottom)[52]. The electronic circular dichroism (ECD) signal of **I** was observed not only in the ultraviolet–vis region, but also in the near-infrared region (Fig. 4b, top). The ECD spectrum observed in the $d \rightarrow d$ transition region demonstrated that the nickel centre in **I** is chiral even in solution. A feature of the observed ECD spectra is that the intensity of the near-infrared bands is significantly stronger than that of the visible absorption. Density functional theory (DFT) calculations on the noncentrosymmetric **I** at the level of UM06/6-311G($d,p$) (SDD for Ni) were also performed (Supplementary Figs 9–14, Supplementary Tables 4 and 5)[54,55]. The simulated infrared and ECD spectra fitted reasonably well with the experimental data. These results suggest that (i) Ni(II)–diamine–acetate **I** retains pseudo-octahedral structure in the solution state, in which the two structurally distinct acetates coordinate to the Ni(II) centre in bidentate manner and (ii) chirality-at-metal in **I** is substitutionally and configurationally inert even in the presence of an excess of coordinative THF.

**A plausible catalytic cycle.** Structural analyses of **I** in the solid and solution states (Figs 2–4), as well as their catalytic activity differences (Table 2), suggested that the monomeric species should predominantly control the stereo-discrimination process (Fig. 5). We assume that the exposed nature of the d-orbital allows it to interact with α-ketoester **1a** as a Lewis acid, making the ketoester susceptible to deprotonation by the acetate ligand acting as a Brønsted base, leading to the formation of (Z)-Ni(II)–enolate that contains the five-membered chelating ring (Fig. 5, step (i)). Based on the X-ray (Fig. 2b) and EDD analyses of **I** (Fig. 3), we propose a Ni(II)–enolate, in which the carbonyl group in the ester in **1** coordinates to Ni(II) at the pseudoapical position. A new perspective in the transient Ni(II)–enolate is its coordination pattern in the octahedral structure; the Ni(II)–enolate reported by Evans[7,8] occupies the same plane with the chiral diamine. The crystallographic evidence that the N–H functionality at the equatorial position in Ni(II) complex **I** can contribute to activating the Lewis base (Fig. 2b) suggests that the subsequent H-bonding activation of the nitrone **2a** (refs 34,44,45) would be a key driving force for fixing the two reaction components in close proximity (Fig. 5, step (ii)), thereby enhancing the reaction rate with high diastereo- and enantioselectivity (Fig. 5, step (iii)). In the proposed model, the centrochiral Ni(II) directly activates the α-ketoester **1a** with the ligand-enabled H-bonding activation of the nitrone **2a**. The model presented herein can explain the obtained absolute stereochemistry of **3aa**.

**Scope and chemoselectivity of Ni(II) catalysis.** The scope of the formal IED [3 + 2] cycloaddition using the catalytic triad of Ni(OAc)$_2$, (R,R)-**4e** and $^i$PrNH$_2$ was examined (Fig. 6). α-Ketoesters **1** bearing various substituents on the aromatic moiety as well as longer alkyl chain substrates served as substrates, giving the corresponding [3 + 2] adducts **3**. Substrate **1h**, bearing the sterically demanding 1-adamantyl group can also participate in the catalytic reaction, affording the corresponding isoxazolidine **3ha** in 74% yield, with 25/1 d.r. and 94% e.e. The terminal olefin in **1i** remains intact in the reaction of **2a**, giving **3ia** in 77% yield, 35/1 d.r. and 88% e.e. Substituted (E)-nitrones (**3ab**, **3ac** and **3ad**) were also applicable in the developed catalytic system, and comparable reactivity and selectivity were obtained by slightly tuning the reaction conditions.

A characteristic feature in this Ni(II)-catalysis is its chemoselective recognition of α-ketoesters and (E)-nitrones. When the reaction was performed using **2a** with vinyl ether **5** instead of **1a**, no reaction took place (Fig. 7a); this is complementary to the reported IED [3 + 2] cycloadditions[25–34], and supports the validity of our working model, which involves a Ni(II)–enolate as the active dipolarophile. The observation that no reaction occurred when we applied the catalytic system to the reaction of (Z)-**6** and **1a** (Fig. 7b) represents additional evidence that the present catalytic system selectively activates (E)-nitrones. Another issue upon which we focused is the use of E/Z-isomerizable nitrone **7** (ref. 56) in our catalytic system (Fig. 7c), because controlling the isomerization is an issue of contemporary interest in organic synthesis[57] as well as in biology[58]. The formal [3 + 2] cycloaddition of E/Z isomerizable ester-conjugated nitrone **7** with **1a** selectively afforded (3S,4R,5R)-anti-anti-**8** as a single diastereoisomer in 90% yield with 64% e.e. The geometry selection of nitrone in the present catalytic system is complementary to that reported in (Z)-nitrone-selective reactions using chiral Cu(II) catalyst[31]. All these considerations indicate that the structure and geometry of both α-ketoester **1** and (E)-nitrone **2** can be discriminated in the present Ni(II)-catalyst system, facilitating the formation of stereochemically complex isoxazolidines **3** and **8**.

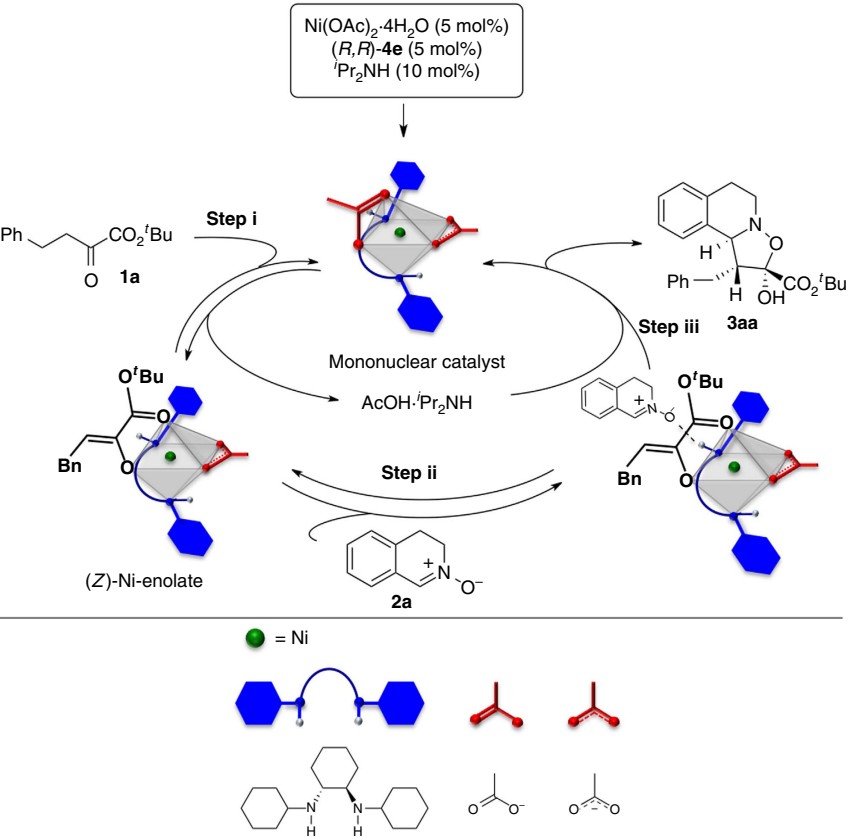

**Figure 5 | Proposed catalytic cycle.** The formal [3 + 2] cycloaddition of **1a** with **2a** using the catalytic triad Ni(OAc)$_2$, (*R,R*)-**4e** and $_i$PrNH2.

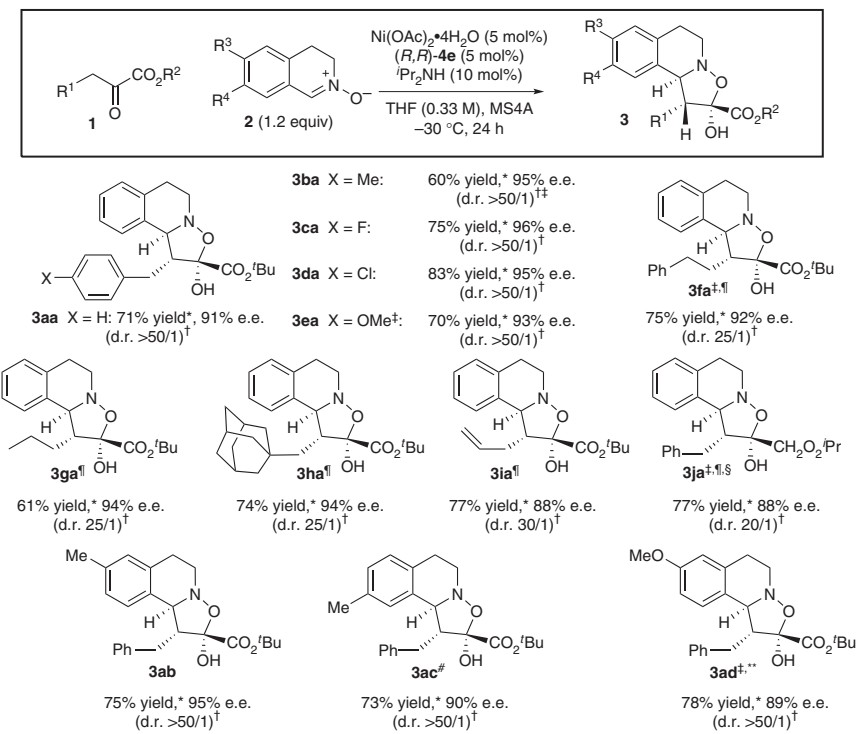

**Figure 6 | Scope of Ni(II)-catalysed [3 + 2] cycloaddition.** *Yields are for isolated materials. †d.r. values were determined from $^1$H NMR spectra of crude mixture. ‡Run for 48 h. ¶2.0 equiv. of **2a** was used. §Run at a concentration of 1.0 M. #10 mol% of metal complex and 3.0 equiv. of **2a** and Et$_3$N (10 mol%) were used. **10 mol% of metal complex and 3.0 equiv. of **2a** were used.

**Figure 7 | Chemoselectivity of Ni(II)-catalysed [3 + 2] cycloaddition.** (**a**) Reaction of vinyl ether **5** with **2a**, (**b**) reaction of **1a** with (Z)-nitrone **6** and (**c**) reaction of **1a** with E/Z-isomerizable nitrone **7**. Reactions were carried out under the conditions shown in the box in Fig. 6.

Here, we not only showcase the first catalytic, asymmetric and direct formal [3 + 2] cycloaddition of transient α-ketoester enolates, but also broaden the concepts underlying the design of asymmetric catalysts. A series of spectroscopic analyses in both the solid and solution states, supported by DFT calculations, were able to relate the electronic structural features of the distorted, desymmetrized Ni(II) complex to its catalytic activity. Specifically, we demonstrated that the EDD analysis of the chiral catalyst is a powerful methodology to experimentally visualize its electron density topology, and provides a quantitative insight into how the catalyst initiates the reaction. The presence of naked d-orbital interacting with the labile acetate ligand allows this acetate ligand to act as a Brønsted base to generate the (Λ)-Ni(II)–enolate. Furthermore, the bifunctional catalytic motif that enables the merger of H-bonding activation with amine ligand and enolate formation provides a mechanistic basis to expand the potential utility of ligand-induced octahedral metal centrochirality[19–23]. The operational simplicity and the easy tunability of the present catalytic system holds a vast potential for designing chiral chemospecific spaces with high predictability.

## Methods

**General.** Detailed experimental procedures, characterization of compounds, Cartesian coordinates and the computational details can be found in Supplementary Figs 1–50, Supplementary Tables 1–5 and Supplementary Methods.

**Preparation of THF solution of the Ni(II) complex.** To a solution of (R,R)-**4e** (266.6 mg, 0.958 mol) in EtOH was added nickel acetate tetrahydrate (237.6 mg, 0.958 mmol) at room temperature. The mixture was stirred for 1.5 h at room temperature, and then filtered through a membrane filter (Chromatodisc 13 N, KURABO). The filtrate was concentrated under reduced pressure to give the Ni(OAc)₂·4H₂O/(R,R)-**4e** = 1/1 complex as a green oil (428.8 mg, 0.813 mmol; calculated on the basis of the formula Ni[(OAc)₂(H₂O)₄(R,R)-**4e**]). Then, THF (4.07 ml) was added at room temperature to prepare the stock solution of Ni(OAc)₂·4H₂O/(R,R)-**4e** = 1/1 (0.2 M in THF). In Fig. 6, this stock solution was used for the catalytic asymmetric formal [3 + 2] cycloadditions of α-ketoesters **1** and nitrones **2**.

**Preparation of mononuclear Ni(II) complex I.** After removal of THF (6.75 ml, 1.35 mmol; calculated on the basis of the formula Ni[(OAc)₂(H₂O)₄(R,R)-**4e**] from the stock solution [Ni(OAc)₂·4H₂O/(R,R)-**4e** = 1/1 (0.2 M in THF)], the residue (692.5 mg) was dissolved in CH₂Cl₂ (∼3 ml) at room temperature. Slow evaporation at −25 °C afforded rich green-colored crystals, which were collected by filtration to give mononuclear Ni(II)–diamine–acetates **I** in 86% yield (586.1 mg, 1.11 mmol; calculated on the basis of the formula Ni[(OAc)₂**4e**(THF)]).

**Preparation of trinuclear Ni(II) complex II.** After removal of THF (2.07 ml, 0.414 mmol; calculated on the basis of the formula Ni[(OAc)₂(H₂O)₄(R,R)-**4e**]) from the stock solution [Ni(OAc)₂·4H₂O/(R,R)-**4e** = 1/1 (0.2 M in THF)], the residue (300.1 mg) was dissolved in n-hexane (∼5 ml) at room temperature. Slow evaporation at room temperature afforded rich green-colored crystals, which were collected by filtration to give trinuclear Ni(II)–diamine–acetates **II** in 85% yield (Ni-based, 182.7 mg, 0.163 mmol; calculated on the basis of the formula Ni₃[(OAc)₆**4e**₂(H₂O)₂]).

**Catalytic formal [3 + 2] cycloaddition of 1a with 2a.** MS 4A (100 mg, powder purchased from Nacalai Tesque) in a Schlenk flask equipped with a magnetic stirring bar was flame-dried under reduced pressure for 5 min. Upon cooling to room temperature, the flask was refilled with N₂, and α-ketoester **1a** (23.4 mg, 0.10 mmol) and (E)-nitrone **2a** (17.7 mg, 0.12 mmol) were added and dried under vacuum. The flask was backfilled with N₂, THF (250 µl) was added at room temperature, and the flask was cooled to –30 °C. To the resulting solution was added the prepared catalyst solution (5 mol%: 25 µl, 0.2 M in THF) and ⁱPr₂NH (10 mol%: 25 µl, 0.4 M in THF). The reaction mixture was stirred for 24 h at –30 °C. Aluminium oxide 60 (∼20 mg, Merck) was added, and the mixture was diluted with EtOAc cooled at –30 °C. The solution was passed through a pad of Aluminium oxide 60, to remove the nickel catalyst, and then eluted with EtOAc and concentrated under reduced pressure. The d.r. (>50/1) was determined from the ¹H NMR spectrum of the crude sample. The residue was purified by column chromatography [CHROMATOREX NH (NH–DM1020, 100–200 mesh, Fuji Silysia Chemical) to give **3aa** in 74% yield (28.2 mg, 0.0740 mmol). The e.e. of (−)-(1R,2R,10bS)-**3aa** (91% e.e.) was determined by means of chiral HPLC analysis (CHIRALCEL OD-H, 0.46 cm (φ) × 25 cm (L), n-hexane/2-propanol = 95/5, 1.0 ml min⁻¹, major; 9.8 min, minor; 17.7 min).

**Electron density distribution analyses of I·THF.** The diffraction data were collected using a RIGAKU AFC-8 diffractometer equipped with a Saturn70 CCD detector with MoKα radiation by an oscillation method at 90 K. X-rays were monochromated and focused by a confocal mirror. Sixteen data sets were measured with different crystal orientations and detector positions. For all data sets, camera distance was 40 mm. Bragg spots were integrated, scaled and averaged up to sin θ/λ = 1.22 Å⁻¹ by the programme HKL2000 (ref. 59) Lorentz and polarization corrections were applied during the scaling processes. Analytical absorption corrections[60] were applied. The initial structure of **I·THF** was solved by a direct method using the programs SIR2004 (ref. 61), and refined by a full matrix least-squares method on F² using the programme SHELXL2014 (ref. 62). Refinements with a multipole expansion method using the Hansen–Coppens multipole formalism[63] and topological analyses based on the resulting parameters were performed with the XD2006 package[64]. Crystal data of **I** for EDD analysis is provided in Supplementary Data 1.

**Data availability.** The X-ray crystallographic data for compounds (1R,2R,10bS)-anti-anti-**3aa**, (3S,4R,5R)-anti-anti-**8**, mononuclear Ni complex **I**, trinuclear Ni complex **II** and multipole population parameters of **I** for electron density distribution analysis have been deposited at the Cambridge Crystallographic Data Centre (CCDC), with the accession codes CCDC 1482737, 1482738, 1482739, 1482740 and 1482741 (http://www.ccdc.cam.ac.uk/data_request/cif). All other data is available from the authors upon reasonable request.

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

## Acknowledgements

This work was supported in part by Scientific Research (C) from the MEXT (24550128) and Project Funding from RIKEN. RIKEN Integrated Cluster of Clusters (RICC) and RIKEN Hokusai GreatWave (GW) provided the computer resources for the DFT calculations.

## Author contributions

Y.S. and D.H. conceived and developed the concept with M.S. who supervised and guide the research. G.N. made the initial discovery of the reaction with S.L. and Y.S. G.N. and Y.S. performed chemical experiments. G.N. prepared the crystals, and D.H. performed all crystallographic studies. Y.S. performed structural characterization of Ni(II) complexes in the solution state with A.M. and M.U. Y.S. performed DFT calculations with A.M. and M.U. The manuscript was written by Y.S. with D.H. and M.S. T.T. is a supervisor of G.N. in Meiji University. All authors discussed the results.

## Additional information

**Competing interests:** The authors declare no competing financial interests.

