## [Peer review file · Nature Communications]

Editorial Note: Parts of this peer review file have been redacted as indicated to maintain the confidentiality of unpublished data. When text is deleted in the peer review file, “[Unpublished data redacted]” has been added in that location.

Reviewers' comments:

Reviewer #1 (Remarks to the Author):

A. Summary of the key results

The crux of the matter in this manuscript is the development of a new set of cycloaddition reactions whereby a transition metal complex based on Ni(II) activated a simple alpha keto ester and the derived enolate participates in an “inverse electron demand” dipolar cycloaddition with cyclic nitrones derived from hydroisoquinolines. The resulting heterocycles contain multiple stereogenic centers and the chiral catalyst delivers the products with good levels of enantiomer selectivity.

B. Originality and interest: if not novel, please give references

This paper builds on the group's prior work using alpha keto esters as pronucleophiles via enolization under exceptionally mild conditions. As opposed to alpha trapping (e.g. halogenation), the cycloaddition mode is new and creates new structures in a straightforward fashion. This paper should be of great interest to those scientists concerned with catalysis, organometallic chemistry, drug discovery (i.e. novel building blocks).

C. Data & methodology: validity of approach, quality of data, quality of presentation

The quality of data throughout is high and the authors use a range of experimental techniques to probe the general reactivity at play and also to understand the structure of the complexes that are doing the catalysis. Relative to “normal” papers in the area of synthesis, this paper uses an entire toolbox to understand what is going on: x-ray diffraction, computations, IR, ECD, etc.

D. Appropriate use of statistics and treatment of uncertainties

n/a

E. Conclusions: robustness, validity, reliability

Appears to be robust based on available data.

F. Suggested improvements: experiments, data for possible revision

The authors have done a good job of being thorough in their experimentation. Given the proposed role of hydrogen bonding in the transition state of the key transformation, I wonder if they have considered what the effect would be by deuterating these sites, either on reaction rate or selectivity.

G. References: appropriate credit to previous work?

Yes

H. Clarity and context: lucidity of abstract/summary, appropriateness of abstract, introduction and conclusions

The “inverse electron demand” descriptor is typically applied to Diels-Alder reactions (i.e. 4+2 cycloadditions). Sustmann has laid out three types of dipolar cycloadditions based on the relevant frontier molecular orbital interactions and the one described in this paper is most accurately described as a Type III dipolar cycloaddition [HOMO(dipolarophile) - LUMO(dipole)]. The paper is well laid out and the work has the strong potential to lay the framework for other catalyst development that takes advantage of the structural phenomena observed here.

Reviewer #2 (Remarks to the Author):

Sohtome and Sodeoka et al. developed Nickel(II)-catalyzed the enantioselective formal [3+2] cycloaddition of 2-oxoesters with cyclic nitrones. The in situ-generated Ni(II) enolates reacted with cyclic nitrones in good yield with high enantioselectivity. This new reaction is attractive because three chiral centers can be introduced to products. However, R1 of 2-oxoesters 1 were limited to primary alkyl, allyl, and benzyl groups. Nitrones 2 were limited to 3,4-dihydroisoquinoline 2-oxides. These narrow substrate scope is not enough synthetically. Synthetic utility of products are not shown here. In contrast, the authors investigated the structure of this active catalyst and the reaction mechanism in detail. Monomeric complex was an active species, and a cooperative mechanism of enolate binding and outer sphere hydrogen-bonding activation was proposed. These results are interesting as new asymmetric catalysis, but the quality is not enough to be published in Nature Communication.

Reviewer #3 (Remarks to the Author):

Sohtome, Hashizume, Sodeoka et al. present a combined synthetic, charge density and DFT study of a genuine at the Ni(II) atom chiral catalyst in asymmetric [3+2] cycloaddition reactions. This is the first time that a concerted approach of so high-calibre analytical methods could be applied to such a hot topic in synthesis. This is an outstanding paper!

The data presented provide much plausibility to the proposed mechanism and should therefore be of both specific and general interest to the synthetic organic and organometallic chemist. It appears to fit perfectly with the remit of Nature Communications, so I am happy to recommend that it is accepted without reservations.

Some questions come to mind on reading this communication, which the authors might want to address in any rewriting:

1. Clearly one of the outstanding early protagonists in enolate chemistry is missing in the bibliography and e. g. Seebach, D. *Angew. Chem.* **1988**, *100*, 1685-1715 should be referenced.

2. To the CD HB article of Lecomte et al. (ref 36) the more recent review from Overgaard and Iversen should be added: *Struct. Bond.* **2012**, *146*, 53-74.

3. Rather than depicting the static deformation density in Fig. 2c, the authors should opt for the Laplacian density distribution. The maximum at the VSCCs at the oxygen atoms should be localised and quantified in $e\text{\AA}^{-5}$. This enables the distinct discussion of geometry and strengths rather than only qualitative hand waving arguments (...directed towards the electron-rich region at Ni(II)...etc.). I suggest to shift the IR and UV spectra in Fig 2 d and e into the SI and replace them by plots of the Laplacian distributions $\nabla^2\rho_{BCP}[e\text{\AA}^{-5}]$ along bond paths $r_{BCP}[\text{\AA}]$ of the six Ni-O(N) bonds.

Minor points:

I believe in line three from the bottom at page 9 Fig. 2b is meant rather than 3b.

Reviewer #4 (Remarks to the Author):

the calculations were a bit boring but well documented.
so the paper is OK.

Editorial Note: Parts of this peer review file have been redacted as indicated to maintain the confidentiality of unpublished data. When text is deleted in the peer review file, “[Unpublished data redacted]” has been added in that location.

Responses to Reviewer

Reviewer 1

F. Suggested improvements: experiments, data for possible revision

The authors have done a good job of being thorough in their experimentation. Given the proposed role of hydrogen bonding in the transition state of the key transformation, I wonder if they have considered what the effect would be by deuterating these sites, either on reaction rate or selectivity.

Response 1

We are very grateful to reviewer 1 for suggesting a possible experiment. To further gain insight into the transition state, the suggested idea sounds very attractive. Indeed, we have also deeply considered it. However, unfortunately, we reached the conclusion that we would be unlikely to be able to discuss the contribution of the N-H functionalities in the diamine ligand by simply deuterating these sites for two main technical reasons.

First, in order to discuss the outer sphere hydrogen-bonding activation using deuterated ligand on the basis of our tactics to link the catalytic activity with electronic structure both in the solid- and solution states, we ideally have to prepare the Ni(II)–THF complex with the deuterated ligand. However, we would have to say it is technically difficult because deuteration of the N-H in the diamine ligand may affect the coordination around the Ni(II) center.

Second, because the stereoselectivity of the developed formal [3+2] cycloaddition of α -ketoesters and (*E*)-nitrones is controlled under proton-transfer conditions, we have to also deuterate the α -position of α -ketoester in order to avoid hydrogen-deuterium exchange in the diamine ligand. However, if we deuterate both the diamine ligand and α -ketoester, we cannot simply discuss the contribution of outer sphere hydrogen-bonding activation to the reaction outcomes (yield, dr and ee).

However, we would like to emphasize that our results on the basis of solid- (Fig. 2 and Fig 3, revised main text) and solution state analyses of the Ni(II) catalyst (Fig. 4, revised main text), together with theoretical calculations (Supplementary Figs. 6–11 and Supplementary Tables 4 and 5), provide convincing evidence to both empirically and computationally highlight the ability of N-H functionality in the metal–amine complex as a proton-donor. The model presented herein can also explain the obtained absolute stereochemistry of **3aa** (as we described in lines 16-17, p12, main text). Considering that only circumstantial evidence has been reported for discussing the role of the N–H group so far in metal amine complexes [see, a recent review: ref 15. Zhao, B. Han, Z. & Ding, K. The N–H functional group in organometallic catalysis. *Angew. Chem. Int. Ed.* **52**, 4744–4788 (2013)], our work has the potential to unlock certain bottlenecks for designing the asymmetric catalyst, and will appeal to a broad readership in the field of chemistry including catalysis, organometallics, and analytical chemistry. Again, we deeply thank reviewer 1 for giving us the chance to reconsider this critical mechanistic

insight. We believe that the deuteration of the diamine ligand may be a useful technique for a different reaction system from proton transfer conditions.

H. Clarity and context: lucidity of abstract/summary, appropriateness of abstract, introduction and conclusions

The “inverse electron demand” descriptor is typically applied to Diels-Alder reactions (i.e. 4+2 cycloadditions). Sustmann has laid out three types of dipolar cycloadditions based on the relevant frontier molecular orbital interactions and the one described in this paper is most accurately described as a Type III dipolar cycloaddition [HOMO(dipolarophile)-LUMO(dipole)].

Response 2

We thank reviewer 1 for pointing out the Sustmann’s significant contributions to classification of the dipolar cycloaddition mode depending on the frontier molecular orbital. We added the sentence describing Sustmann’s classification and cited his review as reference 26 [Sustmann, R. Orbital energy control of cycloaddition reactivity. *Pure. Appl. Chem.* **40**, 569-593 (1974)] in the revised manuscript. Actually, we cited the selected 10 examples of the related [3+2] cycloadditions with electron-rich olefins (references 16-25) to show the difference from newly developed formal [3+2] cycloaddition of α -ketoester enolates and (*E*)-nitrones. Among the papers we listed, we can find the “inverse electron demand” descriptor in 4 papers (16, 21, 22 and 24 in the revised main text). However, there are no papers to follow the type III dipolar cycloaddition [HOMO(dipolarophile)-LUMO(dipole)]. Thus we also keep the “inverse electron demand (IED)” descriptor.

Concomitantly, we have also changed the reference numbers concerning inverse electron-demand [3+2] cycloadditions (refs 16-25) before reference 26.

Revised

This report provides insights into the elusive mechanistic basis for inverse-electron-demand (IED) [3+2] cycloaddition¹⁶⁻²⁵ of α -ketoesters with (*E*)-nitrones (Fig. 1d). The cycloaddition, which is controlled by the LUMO (lowest unoccupied molecular orbital) of the dipole and the HOMO (highest occupied molecular orbital) of the 1,3-dipolarophile, can be also categorized as “type III” according to Sustmann’s classification.²⁶

Reviewer 2

Sohtome and Sodeoka et al. developed Nickel(II)-catalyzed the enantioselective formal [3+2] cycloaddition of 2-oxoesters with cyclic nitrones. The in situ-generated Ni(II) enolates reacted with cyclic nitrones in good yield with high enantioselectivity. This new reaction is attractive because three chiral centers can be introduced to products. However, R1 of 2-oxoesters 1 were limited to primary alkyl, allyl, and benzyl groups. Nitrones 2 were limited to 3,4-dihydroisoquinoline 2-oxides. These narrow substrate scope is not enough synthetically. Synthetic utility of products are not shown here. In contrast, the authors investigated the structure of this active catalyst and the reaction mechanism in detail. Monomeric complex was an active species, and a

cooperative mechanism of enolate binding and outer sphere hydrogen-bonding activation was proposed. These results are interesting as new asymmetric catalysis, but the quality is not enough to be published in Nature Communication.

Response 3: Mechanistic analyses

As we described in the abstract, this work mainly focused on the development of an empirical methodology for providing quantitative insights into how the catalyst initiates the reaction. Both reviewer 1 (perhaps, an expert in catalysis) and reviewer 3 (perhaps, an expert in electron density analysis) gave us very positive comments from synthetic and analytic viewpoints. Specifically, we believe that our solid and solution state analyses of the Ni(II) catalyst, together with theoretical calculations, to link the electronic structure of the centrochiral octahedral Ni(II) complex and its catalytic activity are of high quality and would be of interest to the wide readership of *Nature Communications*.

Response 4: Scope of α -ketoesters

Reviewer 2 commented on the application part that pertains to claims of substrate scope. In terms of α -ketoester scope, we would like to emphasize that this work built on our original work using α -ketoesters as pronucleophiles [See, ref 27 Nakamura, A. Lectard, S. Hashizume, D. Hamashima, Y. & Sodeoka, M. Diastereo- and enantioselective conjugate addition of α -ketoesters to nitroalkenes catalyzed by chiral Ni(OAc)₂ complex under mild conditions. *J. Am. Chem. Soc.* **132**, 4036–4037 (2010)]. Many research groups followed our soft-enolization concept using α -ketoesters in the field of not only catalysis but also total synthesis; the citation number is 71 on December 2016, suggesting a wide range of scope in terms of α -ketoester enolization concept. Even in comparison with other strategies using metal catalysts or organocatalysts, we believe that the α -ketoester scope described in this manuscript is still top-notch. Furthermore, in addition to the availability of the sterically demanding 1-adamantyl group in **3h**, we showcased that the terminal olefin in **1i** remains intact in the developed [3+2] cycloaddition. This is a key result to show the unique chemoselective recognition of the enolate from α -ketoesters (as described in Fig. 7 in the revised main text). [Unpublished data redacted]

Response 5: Scope of 1,3-dipoles

As we discussed in Fig. 7c in the revised main text, our catalytic Ni–diamine–acetate selectively promotes the formal [3+2] cycloaddition of α -ketoester **1a** with (*E*)-**7**, when we used *E/Z* isomerizable nitronone **7**. We believe that the results provide a new perspective for the geometry selection of nitronone, because it is complementary to that reported in (*Z*)-nitronone-selective reactions using chiral Cu(II) catalyst [Jensen, K. B., Hazell, R. G. & K. A. Jørgensen, Copper(II)-bisoxazoline catalyzed asymmetric 1,3-dipolar cycloaddition reactions of nitronones with electron-rich alkenes. *J. Org. Chem.* **64**, 2353–2360 (1999)].

[Unpublished data redacted]

We deeply thank Reviewer 2 for giving us an opportunity to clearly specify our contribution to broadening the substrate scope.

Reviewer 3

Sohtome, Hashizume, Sodeoka et al. present a combined synthetic, charge density and DFT study of a genuine at the Ni(II) atom chiral catalyst in asymmetric [3+2] cycloaddition reactions. This is the first time that a concerted approach of so high-calibre analytical methods could be applied to such a hot topic in synthesis. This is an outstanding paper! The data presented provide much plausibility to the proposed mechanism and should therefore be of both specific and general interest to the synthetic organic and organometallic chemist. It appears to fit perfectly with the remit of Nature Communications, so I am happy to recommend that it is accepted without reservations.

We deeply thank Reviewer 3 for giving us very positive comments. On the basis of requests by reviewer 3, we have made the following modifications.

Clearly one of the outstanding early protagonists in enolate chemistry is missing in the bibliography and e. g. Seebach, D. *Angew. Chem.* 1988, 100, 1685-1715 should be referenced.

Response 6

According to this suggestion, we have added the suggested work as ref 31 in the revised main text [31. Seebach, D. Structure and reactivity of lithium enolates. From pinacolone to selective C-alkylations of peptides. Difficulties and opportunities afforded by complex structures. *Angew. Chem. Int. Ed.* **27**, 1624–1654 (1988)].

To the CD HB article of Lecomte et al. (ref 36) the more recent review from Overgaard and Iversen should be added: *Struct. Bond.* 2012, 146, 53-74.

Response 7

We have replaced the paper concerning charge density methods in the hydrogen bonding as follows.

Original

Espinosa, E., Souhassou, M., Lachekar, H. & Lecomte, C. Topological analysis of the electron density in hydrogen bonds, *Acta Cryst.*, **B55**, 563 (1999).

Revised

Overgaard, J. & Iversen, B. B. Charge density methods in hydrogen bond studies. *Struct. Bond.* 146, 53–74 (Springer, 2012).

Rather than depicting the static deformation density in Fig. 2c, the authors should opt for the Laplacian density distribution. The maximum at the VSCCs at the oxygen atoms should be localized and quantified in $\text{e}\text{\AA}^{-5}$. This enables the distinct discussion of geometry and strengths rather than only qualitative hand waving arguments (... directed towards the electron-rich region at Ni(II)... etc.). I suggest to shift the IR and UV spectra in Fig 2 d and e into the SI and replace them by plots of the Laplacian distributions $\nabla^2\rho_{\text{BCP}} [\text{e}\text{\AA}^{-5}]$ along bond paths $r_{\text{BCP}} [\text{\AA}]$ of the six Ni-O(N) bonds.

Response 8

We deeply thank Reviewer 3 for giving us very constructive comments about our electron density investigations. On the basis of a recent review as well as excellent leading works concerning electron density and chemical bonding, which are cited as references 39, 40 and 41 in the revised manuscript, we have made efforts to improve the quality of this manuscript.

39. Stalke, D. ed. *Stuct. Bond.* 146, 53–74 (Springer, 2012)

40. Kratzert, D., Leusser, D., Holstein, J. J., Dittrich, B., Abersfelder, K., Scheschkewitz, D. & Stalke, D. An Experimental charge density study of two isomers of hexasilabenzene, *Angew. Chem. Int. Ed.* **52**, 4478-4482 (2013)

41. Flierler, U., Burzler, M., Leusser, D., Henn, J., Ott, H., Braunschweig, H. & Stalke, D. Electron-density investigation of metal-metal bonding in the dinuclear “bolyene” complex [$\{\text{Cp}(\text{CO})_2\text{Mn}\}_2(\mu\text{-BtBu})$], *Angew. Chem. Int. Ed.* **47**, 4321-4325 (2008)

Based on Fig. 3 and Fig. 4 described in reference 40, we have tried to illustrate the Laplacian distribution around Ni(II) of **I**. While we investigated several isosurface levels, we found that it is difficult to visualize both nitrogen atoms and oxygen atoms around the Ni(II) center at the same level [See an example below, in which isosurfaces are illustrated at the levels of -45 (blue) and -89 (green) $\text{e} \text{ \AA}^{-5}$].

To further gain insight into the bonding mode around the distorted Ni(II) in **I**, on the basis of the constructive suggestions by Reviewer 3, we have carefully selected and incorporated illustrations, including (a) static deformation density of **I**, (b) on the O(3)–Ni–O(4) plane, Laplacian distribution (c) on the O(1)–Ni–O(2) plane, (d) on the O(3)–Ni–O(4) plane in Fig 3, revised manuscript. The modifications significantly provide deeper insight not only into the electron density topology at the levels of $\text{e} \text{ \AA}^{-3}$ (Fig. 3a and b) but also the spatial arrangement of the valence shell charge concentrations (VSCCs) at the Ni(II) using the plots of Laplacian distribution [$\nabla^2\rho(\mathbf{r})$] along with bond paths (Fig. 3c and d).

According to the suggestions by reviewer 3 as well as MANSUCRIPT CHECKLIST that requests us to fit a column/page including legend, we have separately illustrated “Structural determination of **I** and **II** in the solid state” (Fig. 2), “Electron density distribution map analyses of **I**” (Fig. 3) and “Structural analysis of **I** in solution” (Fig. 4) in the revised manuscript. Because we did not change the illustration and legends in the revised Fig. 2. and Fig. 4, we described only the modifications for Fig. 3 below.

Again, we deeply thank Reviewer 3 for giving us excellent and constructive comments.

Revised: p8-10, main text

Electron density distribution analysis of **I.** To characterize the bonding mode around the distorted Ni(II) in **I**, we performed EDD analysis³⁸ using single-crystal X-ray diffraction data (Fig. 3). A 3D plot of the static deformation density of **I** highlights its valence electron density topology (Fig. 3a). The distribution of d-orbitals along the coordination axes of the Ni(II) center were found as electron-deficient regions. With respect to ligands, the lone pairs of oxygen [O(1), O(2) and O(3)] and nitrogen [N(1) and N(2)] atoms are directed to the electron-deficient regions at Ni(II), showing the conventional character of coordination bonds. In sharp contrast, the lone pair of O(4) is directed towards the electron-rich region at Ni(II). The 2D static model map on the O(3)–Ni–O(4) plane also demonstrates the different bonding character between the Ni–O(3) and the Ni–O(4) bonds (Fig. 3b). We can also discuss the unique bonding nature of Ni–O(4) with maps of Laplacian distribution [$\nabla^2\rho(\mathbf{r})$] of total EDD along bond paths by comparing the $\nabla^2\rho(\mathbf{r})$ distribution in the O(1)–Ni–O(2) plane (Fig. 3c) and the O(3)–Ni–O(4) plane (Fig. 3d)³⁹⁻⁴¹. The valence shell charge concentration (VSCC) region at the Ni(II) center are located between the bond paths of the Ni–O(1) and the Ni–O(2) bonds on the O(1)–Ni–O(2) plane (Fig. 3c). The VSCCs on O(1) and O(2) expand towards the charge-depletion regions around the Ni(II) center along the bond paths. In contrast, the Ni–O(4) bond path goes through inside the VSCC region at O(4) and the VSCC region at the Ni(II) center on the O(3)–Ni–O(4) plane (Fig. 3d). On the other hand, the Ni–O(3) bond shows similar features to the Ni–O(1) and Ni–O(2) bonds. The results described in Fig. 3 represent experimental evidence that a weaker orbital interaction between Ni(II) and O(4) is involved, while electrons are donated from lone pairs in the orbitals on other N and O atoms to the unfilled d-orbital on Ni(II) (see Supplementary Figs. 6–8). The density at the bond critical point (BCP) of the Ni(II)–O(4) bond [$0.260(2) \text{ e } \text{Å}^{-3}$] is remarkably lower than at the other coordination bonds: Ni(II)–O(1); $0.431(2)$, Ni(II)–O(2); $0.480(2)$, Ni(II)–O(3); $0.494(2)$, Ni(II)–N(1); $0.559(3)$, and Ni(II)–N(2); $0.550(3) \text{ e } \text{Å}^{-3}$. Thus, the dissymmetric, distorted octahedral Ni(II)–diamine–acetates **I** possessing an elongated Ni(II)–oxygen bond has d^8 18-electronic configuration with a weak electrostatic interaction with O(4) at the pseudoapical position. The density at the BCP of the N(1)–H \cdots O(5) is $0.084(10) \text{ e } \text{Å}^{-3}$, which fits reasonably with the topological properties [$d(\text{H}\cdots\text{O}): 2.200 \text{ Å}$, $d(\text{N}\cdots\text{O}): 3.155(2) \text{ Å}$, $\alpha(\text{N}–\text{H}\cdots\text{O}): 159.20^\circ$] of the H-bonding⁴².

Figure 3 | Electron density distribution maps of I. (a) 3-Dimensional isosurface static deformation density of I; surfaces drawn at $+0.2 \text{ e } \text{Å}^{-3}$ in green and at $-0.2 \text{ e } \text{Å}^{-3}$ in orange, (b) static model map on the O(3)–Ni–O(4) plane; contours drawn at $0.05 \text{ e } \text{Å}^{-3}$ interval in blue (positive), red (negative) and black (zero) lines, Laplacian distribution of total EDD (c) on the O(1)–Ni–O(2) plane, (d) on the O(3)–Ni–O(4) plane; the blue and red lines denote negative and positive Laplacian contours, respectively. The contours are drawn at $\pm 2 \times 10^n$, $\pm 4 \times 10^n$, $\pm 8 \times 10^n$ (where $n = 0, 1, 2$) $\text{e } \text{Å}^{-5}$. Bond path (BP) and bond critical points (BCPs) are depicted as orange lines and black dots, respectively, in (c) and (d).

For the cross section at the levels of $e \text{ \AA}^{-3}$ (Supplementary Fig. 6: N(1)–Ni–N(2) and Supplementary Fig. 8: O(1)–Ni–O(2)) and $e \text{ \AA}^{-5}$ (Supplementary Fig. 7: N(1)–Ni–N(2)) have been incorporated in the Supplementary Figs.

Revised: p6, Supplementary Information

Supplementary Figure 6 | Static model map on the N(1)–Ni–N(2) plane; contours drawn at $0.05 e \text{ \AA}^{-3}$ interval in blue (positive), red (negative) and black (zero) lines.

Supplementary Figure 7 | Laplacian distribution of total EDD on the N(1)–Ni–N(2) plane. The blue and red lines denote negative and positive Laplacian contours, respectively. The contours are drawn at $\pm 2 \times 10^n$, $\pm 4 \times 10^n$, $\pm 8 \times 10^n$ (where $n = 0, 1, 2$) $e \text{ \AA}^{-5}$. Bond path (BP) and bond critical points (BCPs) are depicted as orange lines and black dots, respectively.

Supplementary Figure 8 | Static model map on the O(1)–Ni–O(2) plane; contours drawn at $0.05 \text{ e } \text{Å}^{-3}$ interval in blue (positive), red (negative) and black (zero) lines.

I believe in line three from the bottom at page 9 Fig. 2b is meant rather than 3b.

Response 9

Yes. That was our typo-error. We have made changed from “Fig 3b” to “Fig 2b”. See below. Thank you very much for carefully reading our manuscript.

Original

The crystallographic evidence that the N–H functionality at the equatorial position in Ni(II) complex **I** can contribute to activating the Lewis base (Fig. 3b) suggests that...

Revised

The crystallographic evidence that the N–H functionality at the equatorial position in Ni(II) complex **I** can contribute to activating the Lewis base (Fig. 2b) suggests that...

REVIEWERS' COMMENTS:

Reviewer #1 (Remarks to the Author):

This revision of a strong paper in the area of catalysis and synthesis clears the bar for publication in my opinion. That said, I will note that I am a bit surprised at the response to my lone experimental suggestion. The authors' concern about the N-D vs. N-H affecting the Ni(II) coordination is surprising to say the least. I see absolutely no issue here, nor do I see a barrier with deuterating the alpha keto ester and assessing the reaction with the N-D catalyst. I would have thought this would be a trivial and potentially information rich experiment. I guess I am unclear about the authors' concerns. Overall, still a very strong paper that should be of use/interest to the community.

Reviewer #3 (Remarks to the Author):

The revisions and amendments supplied by the authors are sufficient and the paper should be published as stands.